# Cytotoxicity of Quantum Dots in Receptor-Mediated Endocytic and Pinocytic Pathways in Yeast

**DOI:** 10.3390/ijms25094714

**Published:** 2024-04-26

**Authors:** Onyinye Okafor, Kyoungtae Kim

**Affiliations:** Department of Biology, Missouri State University, 901 S National, Springfield, MO 65897, USA; odo84s@missouristate.edu

**Keywords:** yeast, quantum dot, toxicity, endocytosis, pinocytosis

## Abstract

Despite the promising applications of the use of quantum dots (QDs) in the biomedical field, the long-lasting effects of QDs on the cell remain poorly understood. To comprehend the mechanisms underlying the toxic effects of QDs in yeast, we characterized defects associated with receptor-mediated endocytosis (RME) as well as pinocytosis using *Saccharomyces cerevisiae* as a model in the presence of cadmium selenide/zinc sulfide (CdSe/ZnS) QDs. Our findings revealed that QDs led to an inefficient RME at the early, intermediate, and late stages of endocytic patch maturation at the endocytic site, with the prolonged lifespan of GFP fused yeast fimbrin (Sac6-GFP), a late marker of endocytosis. The transit of FM1-43, a lipophilic dye from the plasma membrane to the vacuole, was severely retarded in the presence of QDs. Finally, QDs caused an accumulation of monomeric red fluorescent protein fused carbamoyl phosphate synthetase 1 (mRFP-Cps1), a vacuolar lumen marker in the vacuole. In summary, the present study provides novel insights into the possible impact of CdSe/ZnS QDs on the endocytic machinery, enabling a deeper comprehension of QD toxicity.

## 1. Introduction

Quantum dots (QDs) are nano-sized semiconductor crystals well known for their long-lasting fluorescence [1,2,3], tunable optic properties [4,5,6], high quantum yield [7,8,9], and resistance to photobleaching [10,11,12]. As opposed to other engineered nanomaterials, the high sensitivity, optical electro-chemiluminescence (ECL), and long-lasting photochemical properties of QDs have made them highly suitable for applications in bioimaging [13,14,15,16,17], cell tracking [18,19], targeted drug delivery [20,21,22,23,24], diagnostics [25,26,27,28,29], and antimicrobial remedies [4,13,25,30,31]. However, the potential cytotoxicity of the use of QDs is understudied at least due to the lack of understanding of the molecular mechanisms of their trafficking in and out of cells.

Many recent studies have demonstrated adverse effects of QDs on a variety of cellular components and cellular processes [1,25,32,33,34,35]. It was discovered that QDs can enter the cell via various routes of endocytosis including RME and micropinocytosis and negatively affect major organelles such as the Golgi and lysosome [36,37,38,39]. For example, it has been reported that treating cells with QDs has a negative impact on their mitochondrial function, leading to an increase in the level of reactive oxygen species, which causes apoptotic cell death [40,41]. A recent study has shown that QDs can alter protein structure and function by direct interaction [42]. Additionally, a recently published article has proposed that QDs can alter the protein profile of yeast cells [43]. However, the molecular mechanisms behind the QD toxicity presented by the studies above are unclear. Given most studies on investigating the toxic effects of QDs have used metal-based QDs that contain cores including cadmium (Cd) and telluride [44,45,46,47,48,49,50], it had been postulated that metal ions released from the core might account for the observed toxicity in the cell. Indeed, findings from studies have highlighted that the toxicity of QDs stems from the leakage of their core metals [47,51,52,53]. To minimize Cd^2+^ leakage, scientists have added a protective layer of zinc sulfide (ZnS) on the cadmium core. Despite the presence of this coat, the potential QD interaction with structural components in the cell could induce unexpected changes to cells. It is worth noting that this area of research is still in its infancy, but the understanding of the underlying molecular mechanism of QDs’ cytotoxicity in this area is essential before their universal application in medicine.

Although the study of nanomaterials’ impact on animal or human cells/tissues will result in a plethora of new knowledge on their toxic effects, in our research, we selected *Saccharomyces cerevisiae*, commonly known as budding yeast, as the model organism because of its widespread occurrence in natural environments and its significance in industrial applications. Due to the increased accumulation of nanoparticles, including QDs, their negative impacts on the food chain have gained much attention, and therefore, the selection of the budding yeast for the study offers significant insight into the molecular mechanism of QD-mediated toxicity. We utilized carboxylated QDs for the current study. These carboxylated QDs can be used as a drug carrier for efficient delivery of the drug to the target such as human tumors. However, the drug and QDs are known to be separated in the lysosome in the target cells, releasing the QDs in the cytoplasm. In addition, it has been widely demonstrated that QDs enter the cell through many forms of endocytosis, including receptor-mediated endocytosis and micropinocytosis [54,55,56,57,58]. Therefore, QDs can interact with a considerable number of proteins in the cytoplasm and endocytic factors while internalizing. Together, the investigation of the potential impact of carboxylated QDs in the context of endocytosis and vesicular traffic pathways, which are the focus of the current study, is essential to assess any negative impacts of QDs in cells.

Recent RNA sequencing analysis performed in our laboratory identified several differentially expressed genes in response to red CdSe/ZnS-COOH QDs. Particularly, we found that the *APS2* gene, a gene implicated in the endocytic process at the plasma membrane, was downregulated upon the QD treatment [59]. Aps2 is a protein subunit of the AP-2 complex that function primarily in endocytic vesicle maturation and protein sorting at the endocytic sites [60,61,62,63]. This prompted us to propose that the endocytic vesicle or patch maturation at the membrane and the subsequent internalization into the cytoplasm will be negatively impacted by the presence of QDs. However, a previously published study in our laboratory investigated the late stage of endocytosis and found a delayed disassociation of Abp1-GFP (an actin-binding protein of the cortical actin cytoskeleton) from the post-endocytosed vesicle in response to QD treatment [59]. Therefore, one major limitation of that study would be that it focused on just the late endocytic process. As such, this finding motivated our study which delves into the impact of CdSe/ZnS QDs on all three stages: early, middle, and late stages of the endocytic pathways using *Saccharomyces cerevisiae* as an experimental model pathway. Interestingly, several studies have suggested that macropinocytosis is a specific mechanism of QD uptake in mammalian cells [64,65,66]. Therefore, Le et al. explored the partial colocalization of QDs with pinocytosis, indicating that pinocytosis may play a role in QD trafficking [59]. Based on this finding, we proposed that QDs may cause a deceleration in pinocytic traffic toward the vacuole.

Therefore, in the current study, we investigated the influence of CdSe/ZnS QDs on the processes of RME and pinocytic transportation to the vacuole in yeast cells. Using advanced fluorescence microscopy techniques, we tracked the intracellular course of these cellular components in the presence of QDs, and we were able to map out the overall negative impact on these pathways. Overall, our comprehensive study sheds light on novel cellular and molecular mechanisms underlying QD-mediated cytotoxicity in yeast cells, offering valuable insights for future nanotoxicity studies.

## 2. Results

### 2.1. Impact of QDs on the Lifespan of Endocytic Markers

The downregulation of the *APS2* gene in response to the treatment of 25 µg/mL QDs [59] prompted us to determine the turnover rate of early, intermediate, and late stages of endocytic factor recruitment at the endocytic site. Although a yeast growth assay showed that all concentrations of CdSe/ZnS-COOH QDs, ranging from 4 to 50 µg/mL, caused significant reduction in cell growth (Appendix A), most of our research hereafter used 25 µg/mL of QDs to measure the membrane lifespan, cytosolic lifespan, and total lifespan of endocytic patches. Each endocytic marker was fused with GFP to express early (Ede1-GFP), intermediate (Las17-GFP, Sla1-GFP, and Sla2-GFP), and late (Cap1-GFP and Sac6-GFP) endocytic factors. The mean Ede1-GFP lifespan at the membrane in non-treated cells was 36.6 ± 9 s, which was 9.2 s shorter than that in QD-treated cells (Figure 1A,C), suggesting a 25.1 percent increase in Ede1-GFP lifespan with QDs after 6 h. This result indicates that QDs impact on endocytic patch turnover of Ede1, an early endocytic marker. In contrast, our results showed that, 6 h after treatment, the mean membrane lifespan of Las17-GFP in non-treated cells was similar to that in QD-treated cells (Figure 1B,D), suggesting that QDs did not affect Las17-GFP recruitment dynamics.

In the non-treated cells, we determined the Sla2-GFP membrane mean lifespan of 22.0 ± 5.6 s, cytosolic lifespan of 4.5 ± 1.8 s, and total lifespan of 26.7 ± 5.9 s. Upon treatment with QDs, statistically significant increases of 21.3% in membrane lifespan, 22.2% in cytosolic lifespan, and 21.3% in total lifespan were recorded at 6 h post-exposure (Figure 2A,C,D,E). These results imply QDs affect the turnover rate of the Sla2-GFP. However, the mean lifespan of Sla1-GFP at the membrane in non-treated cells was 22.5 ± 4 s, similar to that in QD-treated cells (Figure 2B,F). Sla1 patches are pinched off the plasma membrane and then move toward the cytoplasm [67]. Therefore, we determined the cytosolic mean lifespan of Sla1-GFP after its detachment and found that the time spent in the cytosol was statistically significantly increased in the QD-treated cells (from 2 s to 2.8 s, Figure 2G), a 40 percent increase in Sla1-GFP cytosolic lifetime in QD-treated cells. However, the total mean lifespan of Sla1-GFP at the same focal point in non-treated cells was 24.6 ± 4.3 s, comparable to that (27.9 ± 7.6) in QD-treated cells, with a statistical difference (*p* < 0.05, Figure 3H). Overall, this could imply that QDs cause slow turnover of Sla1-GFP in the cytoplasm but did not affect Sla1 recruitment to the plasma membrane.

The mean lifespan at the plasma membrane and the mean overall lifespan for Cap1-GFP in control cells were 11.0 ± 5.45 s and 14.6 ± 5.37 s, respectively. In contrast, cells treated with QDs exhibited 14.37 ± 4.67 s and 18.3 ± 5.02 s, respectively, increased by 30 percent and 25.2 percent compared to the membrane lifespan and overall lifespan of Cap1-GFP in control cells, respectively (Figure 3A,C,E). A *VPS1* knockout strain expressing Cap1-GFP (Table 1) was also used as a positive control in the experiment. This mutant cell showed a robust increase in Cap1-GFP lifespan at the membrane, over the whole movie duration of 3 min (Appendix A), indicating the loss of *VPS1* further aggravated the defect of Cap1-GFP dynamics in the presence of QDs.

Surprisingly, our results showed that Sac6-GFP, an actin-binding protein at the endocytic site, was associated with the membrane during the movie span of 3 min after 6 h of QD treatment (Figure 3B). This prompted us to determine the specific time at which QDs start to exert adverse effects on the Sac6-GFP lifespan. Our results showed significant changes in Sac6-GFP lifespans at the membrane as early as 5 min after the QD treatment, with an increasing trend of lifespan with longer incubation with QDs (30, 60, and 180 min, Figure 3B,F,G,H). Post-treatment with QDs, the mean membrane lifespans of Sac6-GFP were 10.68 ± 2.54 s, 12.03 ± 5.99 s, 12.27 ± 3.88 s, and 15.26 ± 4.67 at 5 min, 30 min, 1 h, and 3 h. Control cells showed 8.07 ± 1.89 s, 1.13 ± 0.32 s, and 9.2 ± 2.01 s for the mean membrane lifespan, cytosolic lifespan, and total lifespan, respectively. Furthermore, we observed that there was a statistically significant increase in the cytosolic lifespan of Sac6-GFP when cells were incubated with QDs for 30 min, 1 h, and 3 h (Figure 3G). The corresponding cytosolic lifespans of Sac6-GFP at these times were 1.54 ± 0.46 s, 1.89 ± 0.59 s, and 2.38 ± 0.68. Our results showed that the total lifespan of Sac6-GFP (Figure 3H), longer than 30 min with QDs, was statistically significantly increased (Figure 3F). These results signify that Sac6-GFP exhibited an extremely slow turnover rate in the presence of QDs.

### 2.2. Effect of Cadmium-Ion-Mediated Toxicity on Endocytic Marker

Based on current results that revealed a slow turnover rate of Sac6-GFP in the presence of QDs (Figure 3B), we wanted to exclude the possibility that the endocytic delay was not due to the presence of the leaked cadmium ions in the cell. Exposing yeast cells expressing Sac6-GFP to increasing concentrations of cadmium sulfate (CdSO_4_) ranging from 5 to 1000 ppb (1 ppb equals 1 ng/mL), we determined the lifespans of Sac6-GFP between control cells and CdSO_4_-treated cells. As compared to the membrane mean lifespan of Sac6-GFP (8.13 ± 3.75 s) in control, we found statistical differences in the mean membrane lifespan of 50 ppb and above (Figure 4A,B). The mean lifespans at the membrane of 50 ppb, 100 ppb, and 1000 ppb were 11.99 ± 3.98 s, 12.74 ± 5.47 s, and 21.96 ± 8.87 s. Notably, the cytosolic mean lifespan of Sac6-GFP in QD-treated cells showed a 67 percent increase only at the highest concentration of 1000 ppb (Figure 4C). Also, we observed statistical significance in the total mean lifespan of Sac6-GFP at concentrations above 50 ppb (Figure 4D). The total mean lifespan of Sac6-GFP was 13.82 ± 3.98 s, 14.93 ± 5.42 s, and 25.3 ± 9.65 s for 50 ppb, 100 ppb, and 1000 ppb.

In the recent literature, it has been reported that the concentration of Cd^2+^ leakage from QDs fell under the detectible limit of 50 ppb following a 14-day incubation period [42]. Consistently, Cd^2+^ concentration below 50 ppb does not result in any noticeable endocytic defect based on our experiment here (Figure 4), suggesting that all observed endocytic defects (Figure 1, Figure 2 and Figure 3) might be attributed to the presence of the entire structure of QDs, rather than to the leakage of cadmium ions.

### 2.3. QDs Lead to Cps1 Vacuolar Fragmentation Defects

Horstman et al. and Le et al. demonstrated CdSe/ZnS QDs can significantly alter gene expression in yeast cells, particularly genes involved in endocytosis and vesicular transport pathways [59,68]. Particularly, *DID2*, a gene involved in sorting proteins into the multivesicular bodies (MVBs) for transfer into the vacuole, and *COS10*, an endosomal gene predicted to be involved in the MVB sorting pathway, were found to be upregulated upon treatment with QDs [59]. This led us to propose that QDs could affect the integrity of the cargo sorting process at the late endosome. To assess this, we examined the localization of mRFP-Cps1, a type II transmembrane protein sorted to the membrane of the intraluminal vesicles (ILVs) at the late endosome via the MVB pathway [69]. Our results showed that mRFP-Cps1 was properly targeted to the vacuole lumen in NTCs (Figure 5A). However, an increased concentration of QDs (50 µg/mL of green CdSe/ZnS QDs) caused an unexpected severe fragmentation of the vacuole as compared to the NTCs and QD-treated cells (25 µg/mL of green CdSe/ZnS QDs, Figure 5B). Most cells, including the non-treated and treated with 25 µg/mL of QDs groups, exhibited fewer than three vacuoles. In contrast, most cells treated with 50 µg/mL of QDs contained more than three vacuoles. Furthermore, we observed a statistical significance in the mean fluorescence intensity of mRFP-Cps1 based on a line intensity analysis after 6 h of incubation in cells treated with 50 µg/mL of QDs compared to both non-treated cells and cells treated with 25 µg/mL of QDs (Figure 5C), suggesting an accumulation of mRFP-Cps1 in the vacuoles.

### 2.4. QDs Cause Lipophilic FM1-43 Transport Defects

Le et al. revealed that not only receptor-mediated endocytosis but also the pinocytosis route is used for QD trafficking [59]. To test whether the interaction of QDs with the pinocytic pathway leads to the alteration of the pinocytic rate, we traced the traffic of FM1-43 dye, a lipophilic dye that transits from the plasma membrane to the vacuolar membrane [70]. Non-treated cells displayed higher FM1-43 fluorescence intensity at the rim of the vacuole after 6 h of incubation as compared to the QD-treated cells (50 µg/mL of CdSe/ZnS QDs), signifying that the transit to the vacuole is efficient. In the QD-treated cells (50 µg/mL of CdSe/ZnS QDs), FM1-43 intensity was reduced (**** *p* < 0.0001, Figure 6A,B), indicating that the transit rate was significantly delayed.

## 3. Discussion

Previously, several studies have explored the uptake and distribution of QDs in mammalian cells [71,72,73,74,75], however, a thorough examination to reveal the toxic impact of CdSe/ZnS QDs on each stage of endocytosis has not been conducted. Results from this study provide novel insights into the effect of CdSe/ZnS QDs on the various stages of receptor-mediated endocytic and pinocytic pathways in *Saccharomyces cerevisiae*. Accordingly, we propose a model that demonstrates the impact of QDs on each stage of receptor-mediated endocytosis (RME) and pinocytosis (Figure 7A,B). The essence of the model is that QDs slow down the turnover rate of Ede1, Sla2, Cap1, and Sac6 at their corresponding recruitment site or endocytic site as well as their dissociation from the post-internalized vesicle in the cytoplasm (Figure 7B).

Notably, our observation showed that QDs caused a significantly increased lifespan of Sac6-GFP at the endocytic site (at least 6 s to 180 s increases depending on QD concentrations) in comparison to the modestly increased lifespan (~3 s) of Cap1-GFP in response to QDs. This observation is intriguing and provides new insights into the potential interaction between QDs and different endocytic proteins at the endocytic site. Although both Cap1 and Sac6 were not previously detected as a binding partner of QDs, based on the shotgun proteomic analysis [42], we cannot exclude the possibility that Cap1 and Sac6 bind QDs directly or indirectly. It can be proposed that Sac6-GFP binds QDs with higher affinity than Cap1-GFP does. Although this notion should be tested, under this scenario, Sac6-GFP QD complexes at the endocytic site would not only abolish the activity of Sac6 but impede other actin-binding proteins nearby, which together inhibit actin-cytoskeleton-assisted vesicles’ scission at the membrane. To further elucidate the precise nature of the interactions and the binding affinities of QDs with endocytic proteins, including Cap1 and Sac6, further investigative work is warranted.

Le et al. recently reported that treatment with CdSe/ZnS QDs leads to the overexpression of *COS10* and *DID2* genes that are implicated in cargo sorting at the MVB in yeast [59]. Considering the findings of the accumulation of mRFP-Cps1 in the vacuolar lumen, we propose two models that explain the potential molecular mechanisms behind the observed phenomenon. First, it can be postulated that simply overexpression of *COS10* and *DID2* might promote, in the presence of QDs, the sorting of mRFP-Cps1 at the late endosome and the subsequent delivery of it to the vacuole. Secondly, given the downregulation of vacuolar-type H^+^-ATPase (V-ATPase) pumps in response to QDs as described by Le et al. (2023) [59], the accumulation of mRFP-Cps1 might be due to a weak acidity in the vacuole, leading to impaired vesicular degradation of mRFP-Cps1.

Sravya et al. explored the effects of QD exposure on the yeast pinocytic process and found that the transit of FM4-64, a lipophilic dye, was significantly delayed [76]. This finding is consistent with our observation using FM1-43 as a pinocytic marker. Although both studies used CdSe/ZnS QDs with carboxylic ligands from NN-Lab, Sravya et al. employed QDs measuring 7.2 nm in diameter, whereas our research utilized larger QDs with a diameter of 9 nm. Studies have shown that the toxicity of QDs is size dependent, with smaller QDs exhibiting greater toxicity [77,78]. However, based on the evidence from this study as well as that of Sravya et al., it can be deduced that the cytotoxic effects of QDs observed within the pinocytic route are independent of the size, color, and emission of the QD involved.

In conclusion, findings from our study provided evidence that regardless of the specific endocytic pathway utilized by cells, the toxicity of QDs is more plausibly attributed to their entire structure rather than to the release of Cd^2+^ ions from the QDs. Additionally, QDs negatively impacted all stages of receptor-mediated endocytosis. Lastly, the transit of FM1-43 dye via the pinocytic pathway was compromised, and the MVB sorting process was altered in the presence of QDs. These results collectively advance our understanding of QD cytotoxicity at the level of endocytosis. However, the precise mechanism of QD binding to endocytic factors implicated in either RME or pinocytosis should be addressed in future studies.

## 4. Materials and Methods

### 4.1. Yeast Strain and Culturing

Yeast strains utilized in this investigation are listed in Table 1. Strains were streaked on selective agar plates, including yeast peptone dextrose (YPD), or synthetic defined (SD) medium lacking histidine (SD-His). Plates were incubated at 30 °C in a stationary incubator for two to three days until colonies were well established. For liquid cultures, the same media compositions were used, omitting the agar. Prior to each experiment, a single colony from each selective plate was inoculated into 3 mL of the corresponding liquid medium and cultured in a shaker at 30 °C for 24 h to establish a fresh culture. The optical density at 600 nm (OD600) was measured, and the culture was diluted to an OD600 of 0.1. The resulting cultures were then incubated for 6 h at 30 °C with or without the addition of CdSe/ZnS-COOH (QDs) or cadmium sulfate to assess their effects on yeast growth.

### 4.2. Characterization of CdSe/ZnS QDs

Several studies have characterized CdSe/ZnS QDs using various methods such as scanning transmission electron microscopy (STEM), energy dispersive X-ray spectroscopy (EDAX), ultraviolet–visible absorption spectroscopy (UV/Vis), dynamic light scattering (DLS), and photoelectron spectroscopy (XPS), and the results are consistent across all studies [79,80,81,82,83]. In this investigation, we utilized two variants of CdSe/ZnS QDs emitting distinct wavelengths (Table 2), sourced from NN Labs (Fayetteville, AR, USA), as indicated by their catalog numbers (CZW-R-5 and CZW-G-5). These QDs are composed of a CdSe/ZnS core–shell conformation with carboxylic acid (COOH, <% organic impurities, not including ligands) suspended in water (1 mg/1 mL). The red-emitting QDs (catalog # CZW-R-5) measure 9 nm (NN Labs) with emission peaks from 610–620 nm (Appendix A). Concurrently, the green-emitting QDs (catalog # CZW-G-5) are 6.1 nm in size (NN Labs), with emission peaks from 540–560 nm (Appendix A). Previously, Hens et al. characterized green CdSe/ZnS QDs using DLS, STEM, EDAX, and UV/Vis, although they did not establish the emission spectrum of the QDs [46]. We utilized DLS to ascertain the hydrodynamic diameter of the QDs in aqueous dispersion, with red QDs presenting a diameter from 12–13 nm (Appendix A) and green QDs with a diameter from 13–14 nm (Appendix A). Zhang et al. contributed SEM and XPS data, allowing us to visualize and analyze the quantum dots’ morphology and chemical composition [75]. Red CdSe/ZnS QDs were used for all the experiments except for the quantification of mRFP-Cps1 vacuole organization which utilized green CdSe/ZnS QDs.

### 4.3. Assessment of Endocytic Markers’ Recruitment Dynamics

A yeast strain expressing Ede1-GFP, Las17-GFP, Sla1-GFP, Sla2-GFP, Cap1-GFP, or Sac6-GFP (Table 1) was treated with 25 µg/mL of CdSe/ZnS QDs and incubated for 6 h at 30 °C. The treated cells were observed using an Olympus IX81 inverted microscope, equipped with a spinning confocal box (CSU-X1, Yokogawa, Japan) and an ImagEM camera (X2 EM-CCD, Hamamatsu, Japan). Two excitation laser lines, 488 nm and 561 nm were employed for imaging the green and red channels, respectively. Time-lapse videos were captured for 3 min at a frame rate of 3 images per second, resulting in 360 images. Each image had an exposure time of 200 ms. The lifespan of an endocytic marker at the membrane was determined by analyzing these videos. This lifespan was defined as the duration from the marker’s appearance to the point until it either moved away from its origin or disappeared, while the cytosolic lifetime represents the average duration that endocytic markers are present within the cytosol and are no longer visible. A total of forty-five (*n* = 45) patches from three videos for each strain were analyzed to calculate the average patch lifespan at the membrane. A two-sample Student’s *t*-test was conducted in GraphPad Prism 9 to determine if there was a significant difference in the average lifespan, cytosolic lifetime, and total (membrane + cytosolic) lifetime for each yeast strain. More details about this analysis can be found in Section 4.7.

### 4.4. Assessment of Intact Cadmium-Ion-Mediated Toxicity

We employed KKY 0030 to investigate the effects of varying concentrations of cadmium sulfate ions (5 ppb, 25 ppb, 50 ppb, 100 ppb, and 1000 ppb) for a duration of 6 h at 30 °C in a shaker incubator. Following incubation, the yeast cells were placed under a spinning confocal microscope with excitation laser lines of 488 nm and 561 nm for green and red channel imaging, respectively, for creating time-lapse videos as stated in Section 4.3. To create kymograph images the recorded videos were exported to ImageJ version 2.9.0. After selecting an endocytic patch, the multiple kymograph function in ImageJ was used to produce a kymograph image. The membrane lifespan and cytosolic lifetime of each patch were recorded and exported to GraphPad Prism version 9 for further analysis. To determine if there were significant differences in the average lifespan at the membrane and cytosol of each cell, we conducted a non-parametric ANOVA using GraphPad Prism version 9 (see Section 4.7).

### 4.5. Quantification of mRFP-Cps1 Vacuole Organization

The yeast strain (KKY 1494) carrying the mRFP-Cps1 plasmid [84] was exposed to either 25 µg/mL or 50 µg/mL of green CdSe/ZnS-COOH QDs or left untreated for 6 h at 30 °C. The resulting cells were visualized using a confocal microscope. The percentage of cells with fewer than three vacuoles and the peak line intensity in the presence and absence of QDs were determined using a total of sixty (*n* = 60) small, budded cells from each treatment group. Briefly, peak line intensity was analyzed to determine the fluorescence intensity of mRFP-Cps1 using a Slidebook (v6). The differences between the untreated and treated groups were statistically analyzed using the ANOVA non-parametric test in the GraphPad Prism version 9 (see Section 4.7).

### 4.6. Pinocytosis Assay Using FM1-43

Wildtype yeast cells (KKY 0002) (Table 1) were incubated with two different concentrations of QDs, 25 µg/mL and 50 µg/mL, for 6 h at 30 °C in a shaker. After incubation, the cells were grown for an additional 24 h at 30 °C to create a fresh stock. The cells were diluted to achieve an OD of 0.1 at the start of the experiment (0 h) and were then cultured for 6 h in the presence or absence of red CdSe/ZnS-COOH QDs or cadmium sulfate. The cell cultures were centrifuged for 1 min at 2000 rpm at 4 °C. The supernatant was discarded, and the cell pellet was resuspended in 0.5 mL of SD media and centrifuged again. The final cell pellet was resuspended in ice-cold SD media. For the pinocytosis assay, the resuspended cells were stained with FM1-43 dye, a marker for pinocytosis [85], at a final concentration of 500 µM. The uptake of FM1-43 dye by the yeast cells was monitored using fluorescence microscopy using the green channel with an exposure time of 200 ms and a magnification of 1000× oil immersion to determine the impact of CdSe/ZnS QDs on pinocytosis in yeast cells. FM1-43 intensity at the vacuole in the presence and absence of QDs was quantified by counting 120 small, budded cells in each treatment. The data were then analyzed using the ANOVA non-parametric tab in the GraphPad program to determine any significant differences between non-treated and the two groups of treated samples (see Section 4.7).

### 4.7. Statistical Analysis

The statistical analysis across various sections of the study employed a two-tailed, non-parametric approach. For endocytic marker analysis in Section 4.3, the student’s *t*-test was used to compare non-treated vs. treated samples, with results displayed on Prism graphs, each representing an average of forty-five patches with standard deviation and error bars. The statistical significance of data in the graph is indicated by * *p* < 0.05, ** *p* < 0.01, *** *p* < 0.001, and **** *p* < 0.0001. Cadmium ion toxicity assessment, vacuole organization quantification, and pinocytosis assay (in Section 4.4, Section 4.5, and Section 4.6, respectively) were analyzed using GraphPad with a one-way ANOVA and Dunnett’s multiple comparison test to assess variance between groups. Results are represented graphically with the same significance markers as noted earlier. The statistical significance of data in the graph is indicated by * *p* < 0.05, ** *p* < 0.01, *** *p* < 0.001, and **** *p* < 0.0001.

## Figures and Tables

**Figure 1 ijms-25-04714-f001:**
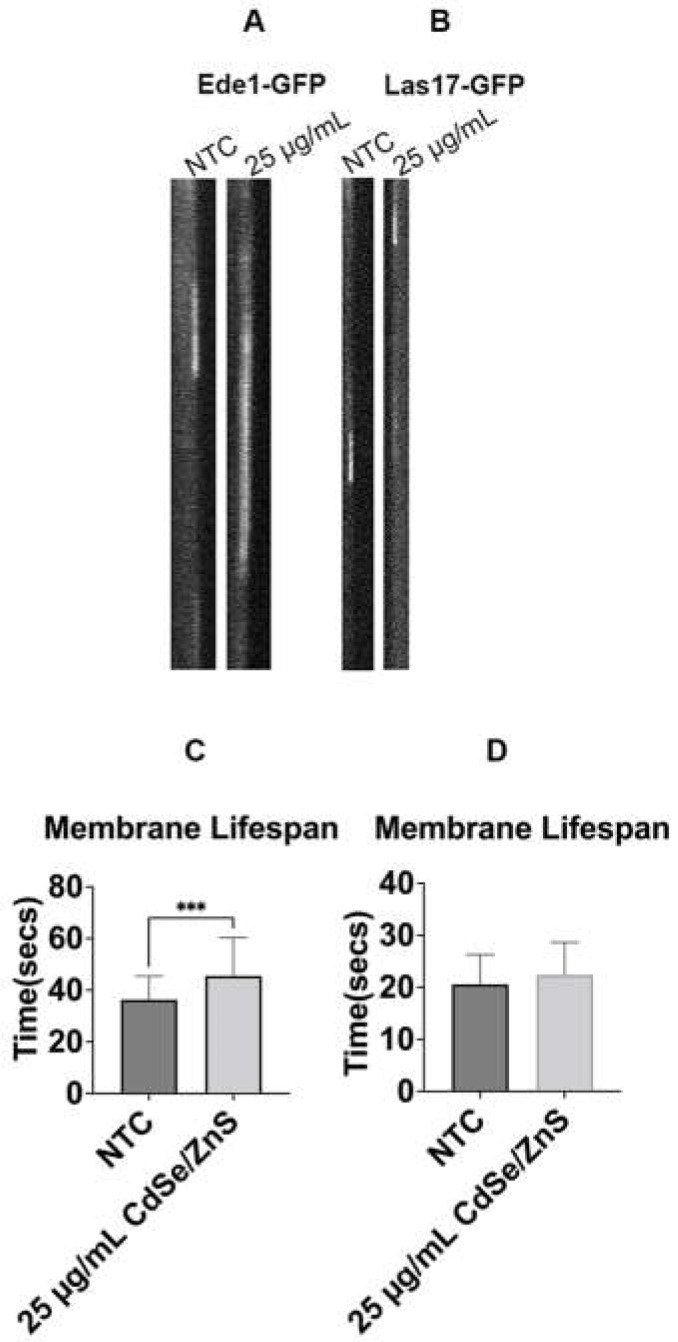
The average membrane lifespans of Ede1-GFP and Las17-GFP patches at the plasma membrane. (**A**) Kymographs of an endocytic patch carrying Ede1-GFP. (**B**) Kymographs of an endocytic vesicle with Las17-GFP. (**C**) Ede1-GFP membrane mean lifespan at 6 h. It shows the average lifespan of Ede1-GFP patches with or without QDs (NTC) in the culture media (25 µg/mL of CdSe/ZnS). (**D**) Las17-GFP lifespan at 6 h. NTC: non-treated cells. *** *p* < 0.001.

**Figure 2 ijms-25-04714-f002:**
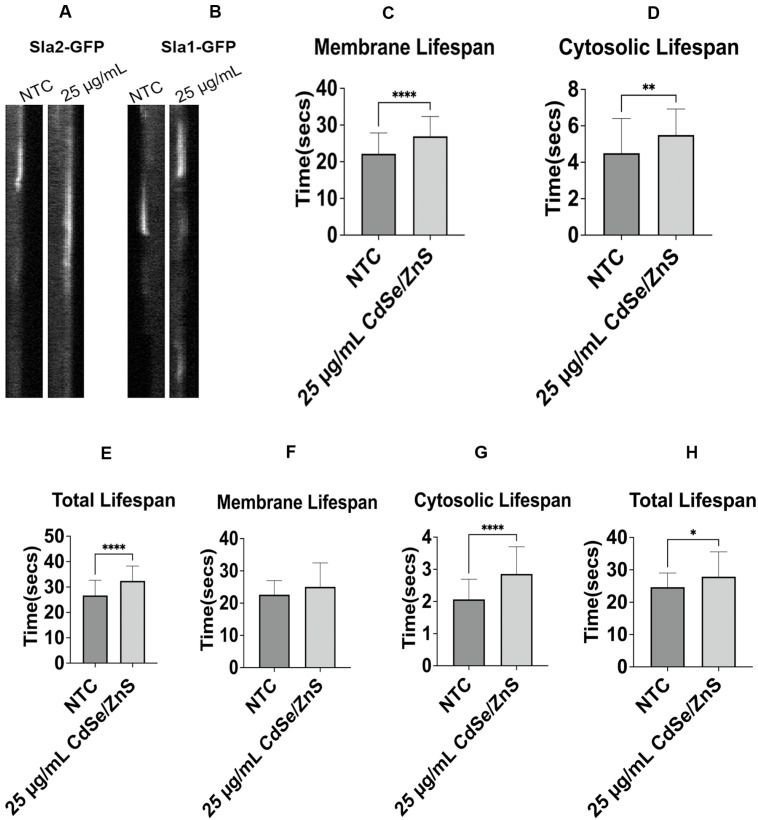
The dynamics of recruitment and dissociation of Sla1 and Sla2 from the endocytic patch. (**A**) Kymographs of an endocytic patch carrying Sla2-GFP. (**B**) Kymographs of an endocytic patch carrying Sla1-GFP. (**C**) Sla2-GFP membrane mean lifespan at 6 h. It shows the average lifespan of Sla2-GFP patches with or without QDs (NTC) in the culture media (25 µg/mL of CdSe/ZnS). (**D**) Sla2-GFP cytosolic mean lifespan at 6 h. (**E**) Sla2-GFP total mean lifespan at 6 h. (**F**) Sla1-GFP membrane mean lifespan at 6 h. (**G**) Sla1-GFP cytosolic mean lifespan at 6 h. (**H**) Sla1-GFP total mean lifespan at 6 h. NTC: non-treated cells. * *p* < 0.05, ** *p* < 0.01, **** *p* < 0.0001.

**Figure 3 ijms-25-04714-f003:**
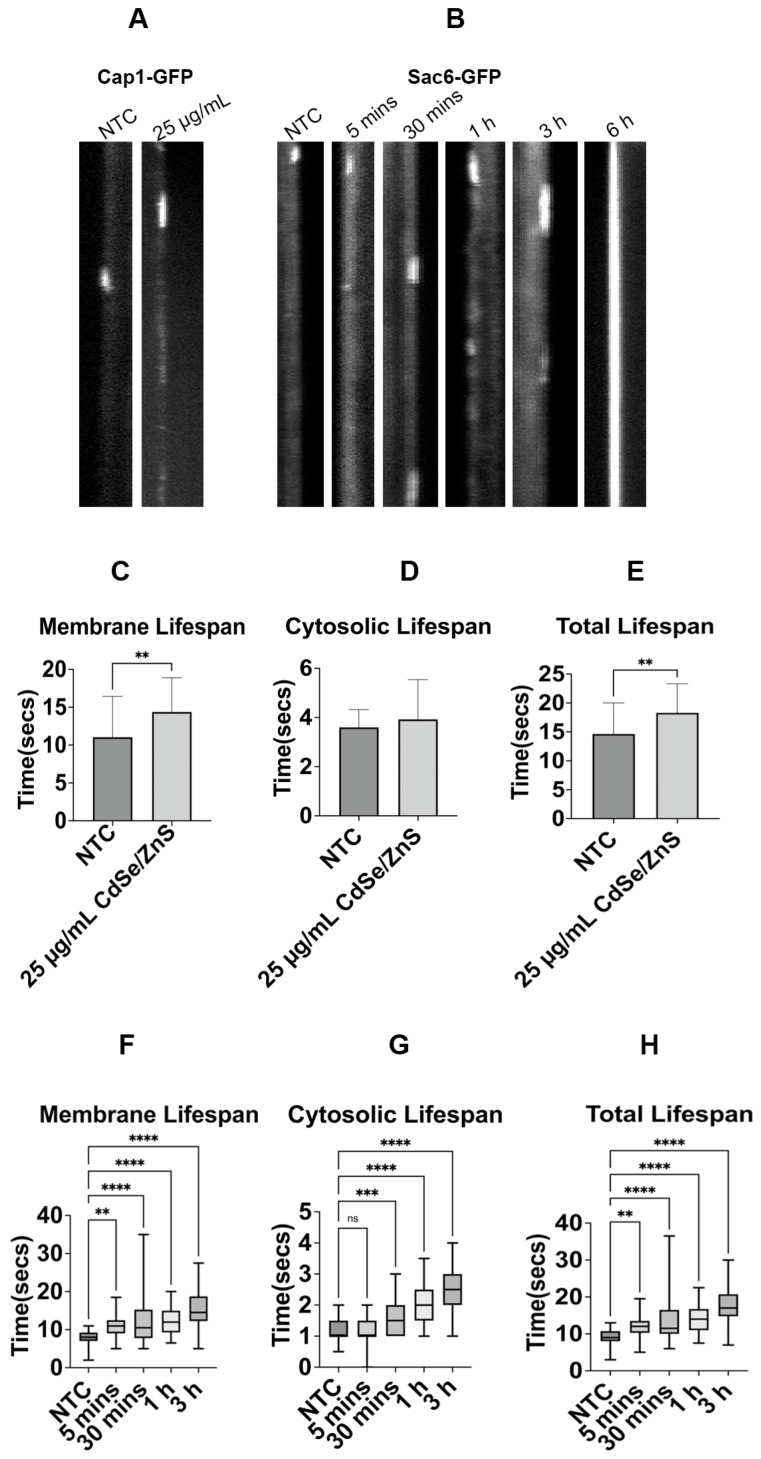
The recruitment dynamics of Cap1-GFP and Sac6-GFP from the endocytic site. (**A**) Kymographs of endocytic patch carrying Cap1-GFP. (**B**) Kymographs of endocytic patch carrying Sac6-GFP. (**C**) Cap1-GFP membrane mean lifespan at 6 h. It shows the average lifespan of Cap1-GFP patches with or without QDs (NTC) in the culture media (25 µg/mL of CdSe/ZnS). (**D**) Cap1-GFP cytosolic mean lifespan at 6 h. (**E**) Cap1-GFP total mean lifespan at 6 h. (**F**) Sac6-GFP membrane mean lifespan at 0, 5 min, 30 min, 1 h, and 3 h. (**G**) Sac6-GFP cytosolic mean lifespan at 0, 5 min, 30 min, 1 h, and 3 h. (**H**) Sac6-GFP total mean lifespan at 0, 5 min, 30 min, 1 h, and 3 h. NTC: non-treated cells. ns—not significant, ** *p* < 0.01, *** *p* < 0.001, **** *p* < 0.0001.

**Figure 4 ijms-25-04714-f004:**
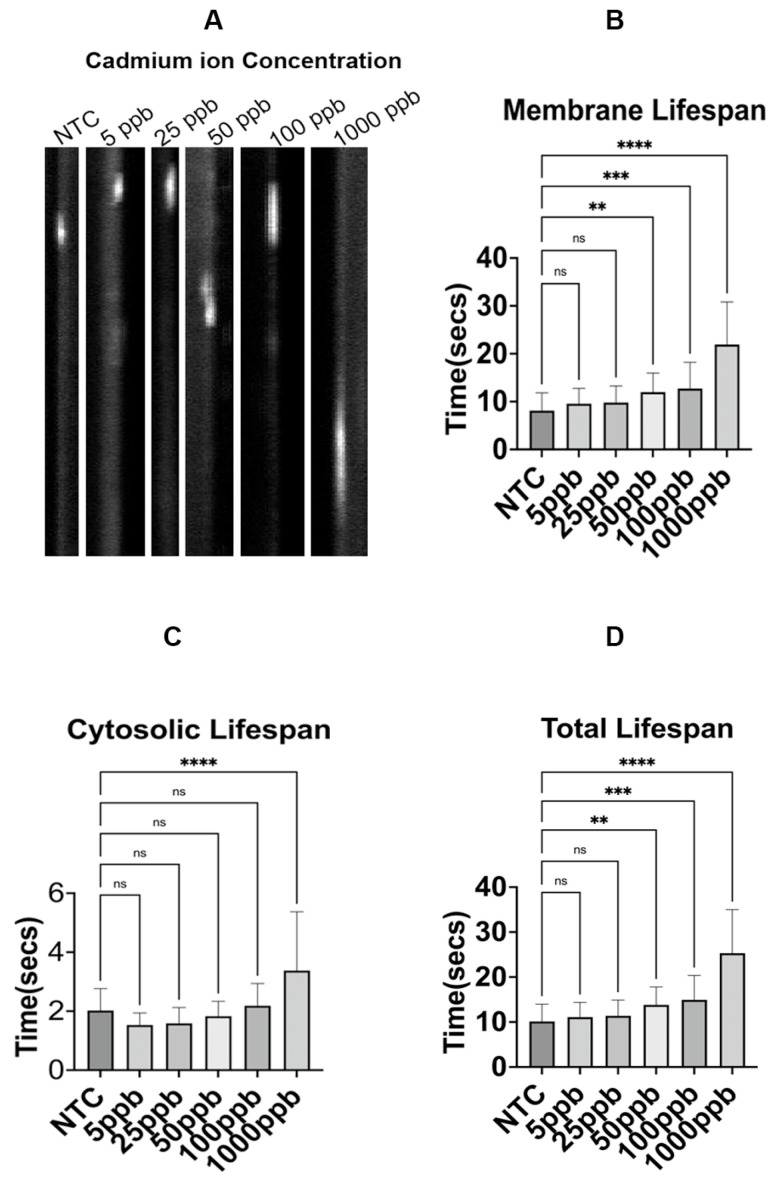
The mean membrane lifespans of Sac6-GFP patches are in response to varying concentrations of cadmium sulfate ions. (**A**) Representative kymographs of endocytic patch carrying Sac6-GFP. (**B**) Mean membrane lifespan of Sac6-GFP with CdSO_4_ at 6 h. It shows the average lifespan of Sac6-GFP patches with or without CdSO_4_ in the culture media (5 ppb, 25 ppb, 50 ppb, 100 ppb, and 1000 ppb of CdSO_4_). (**C**) The mean cytosolic lifespan of Sac6-GFP with CdSO_4_ at 6 h. (**D**) The mean total lifespan of Sac6-GFP with CdSO_4_ at 6 h. NTC: non-treated cells. ns—not significant, ** *p* < 0.01, *** *p* < 0.001, **** *p* < 0.0001.

**Figure 5 ijms-25-04714-f005:**
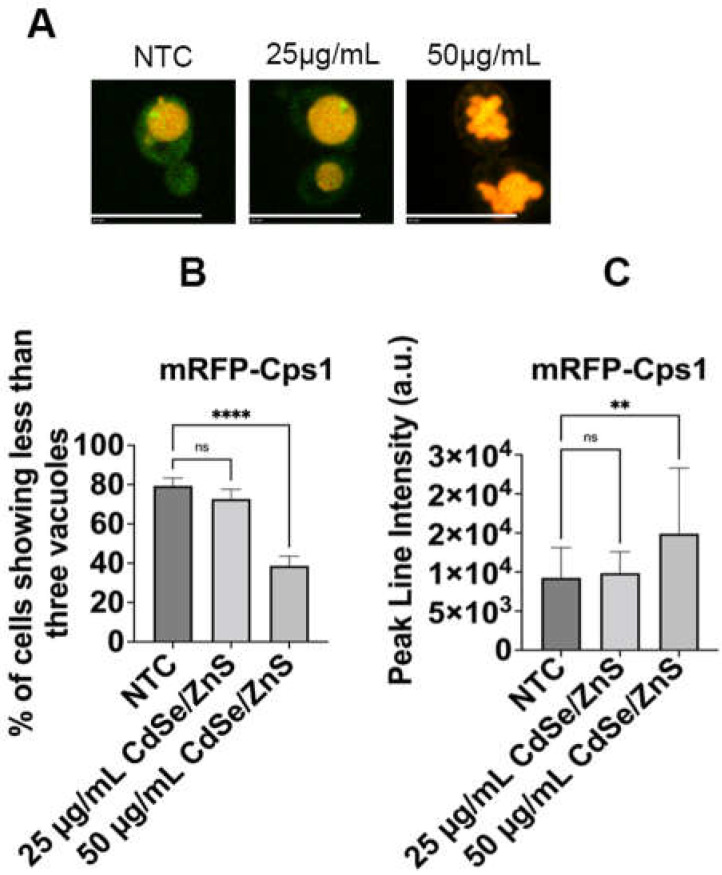
mRFP-Cps1 fragmentation defects with and without the presence of CdSe/ZnS QDs. (**A**) Representative image from NTCs, cells treated with 25 µg/mL of CdSe/ZnS QDs, or cells treated with 50 µg/mL of CdSe/ZnS QDs. The size bar is equivalent to 10 µm. (**B**) Quantification of percentage of cells containing fewer than three (3) vacuoles. (**C**) mRFP-Cps1 peak line intensity. It shows the average mRFP peak line intensity between the non-treated and treated cells (25 µg/mL and 50 µg/mL of CdSe/ZnS QDs treatment). NTC: non-treated cells. ns—not significant, ** *p* < 0.01, **** *p* < 0.0001.

**Figure 6 ijms-25-04714-f006:**
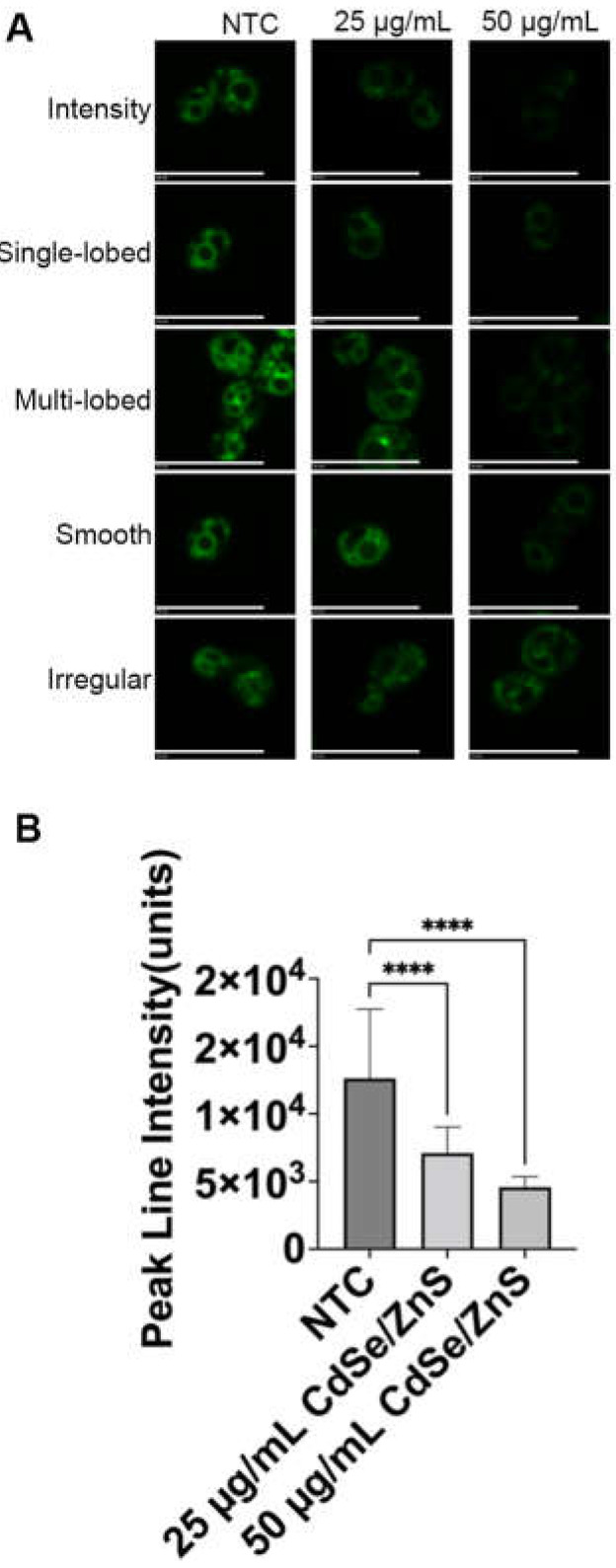
Vacuolar trafficking defect. (**A**) Images of FM1-43 localization in NTC and QD-treated cells. The size bar is equivalent to 10 µm. (**B**) Peak line intensity of FM1-43 at 6 h post-QD treatment with 25 µg/mL and 50 µg/mL of CdSe/ZnS QDs. Dotted circles represent where the cells are concentrated. NTC: non-treated cells. **** *p* < 0.0001.

**Figure 7 ijms-25-04714-f007:**
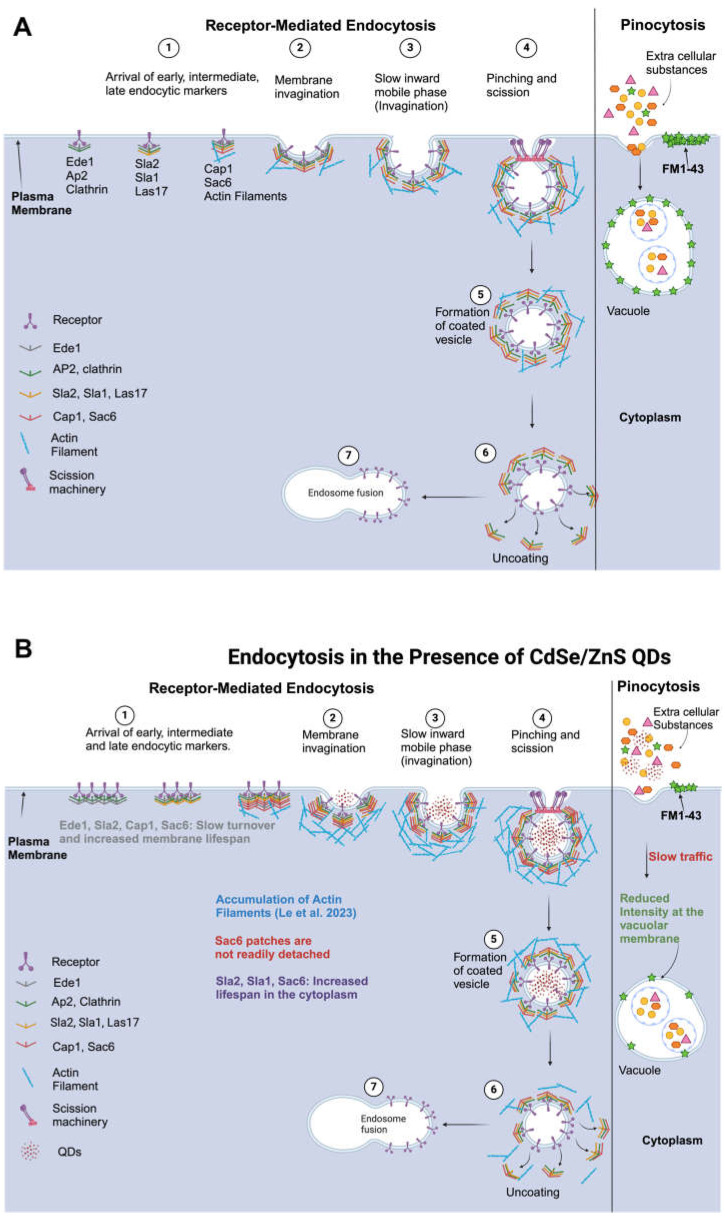
Model showing the effects of QDs on stages of receptor-mediated endocytosis and pinocytosis. (**A**) Receptor-mediated endocytosis and pinocytosis pathway in non-treated WT cells. (**B**) Receptor-mediated endocytosis and pinocytosis pathways in the presence of CdSe/ZnS QDs. Created with BioRender.com (accessed on 24 February 2024).

**Table 1 ijms-25-04714-t001:** Yeast strains used in the current study.

Strain Name	Strain Number	Genotype
Wildtype yeast (BY4741)	KKY 0002	*MATa his3∆1 leu2Δ0 met15∆0 ura3∆0*
Ede1-GFP	KKY 0200	*MATa his3∆1 leu2∆ met15∆ ura3∆ EDE1-GFP-HISMx6*
Las17-GFP	KKY 0093	*MATa his3∆1 leu2∆ met15∆ ura3∆ LAS17-GFP-HIS3*
Sla1-GFP	KKY 0032	*MATa SLA1-GFP-HIS3 his3∆1 leu2∆ ura3∆ lys2∆*
Sla2-GFP	KKY 0254	*MATa his3∆1 leu2∆ ura3∆ lys2∆ SLA2-GFP-HIS*
Cap1-GFP	KKY 0003	*MATa CAP1-GFP-HIS3 his3∆1 leu2∆ met15∆ ura3∆*
Sac6-GFP	KKY 0030	*MATa SAC6-GFP-HIS3 his3∆1 leu2∆ ura3∆ lys2∆*
mRFP-Cps1	KKY 1494	*MATa his3∆1 leu2∆ met15∆ ura3∆ mRFP-Cps1-URA*
Vps1∆ + Cap1-GFP	KKY 0219	*MATa his3∆1 leu2∆ met15∆ ura3∆ CAP1-GFP-HIS3/his3∆1 leu2∆ lys2∆ ura3∆ VPS1:KanMX6*

**Table 2 ijms-25-04714-t002:** Characterization of CdSe/ZnS-COOH QDs.

QD	Emission Color	Catalog # (NN Labs)	Size (nm) Data Provided by NN Labs	SEM-Study-Based Size Measurement	Components Based on XPS
CdSe/ZnS-COOH	Red	CZW-R-5	9	5–10 [75]	Cd3d CdSe, Zn2p ZnS [75]
CdSe/ZnS-COOH	Green	CZW-G-5	6.1	6–9.1 [46]	Cd3d CdSe, Zn2p ZnS [75]

## Data Availability

Data are available upon request.

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
