# Peer review of "Cytotoxicity of Quantum Dots in Receptor-Mediated Endocytic and Pinocytic Pathways in Yeast"

_ijms, 2024, doi:10.3390/ijms25094714_

Round 1
Reviewer 1 Report
Comments and Suggestions for Authors
The presented manuscript by Okafor and Kim focuses on the cytotoxic evaluation of quantum dots' influence on endocytosis and pinocytosis. These processes are of key importance for every cell, the Authors studied them using a yeast cell model. The design of the study is interesting, as well as using the model yeast strains expressing the GFP-tagged proteins of endocytic machinery. However, the manuscript has some flaws that must be corrected before acceptance for publication. Detailed comments are listed below.
Major remark:
Methodology, general - it would be beneficial to compare the results concerning endocytic markers and other molecules dynamics not only with untreated cells but also with some additional positive control (e.g. compounds known to interfere with endocytic pathway)
Minor remarks:
1. Abstract - the abbreviations used (Sca6, FM1-43, mRFP-Cps1), are not commonly recognized ones, so they have to be explained in the abstract.
2. lines 51, 52, 58 - abbreviations used for the first time should be explained
3. Results, lines 181-182 - it would be beneficial to measure the concentration of leaking Cd ions using the tested samples of QDs, just to be sure that the concentrations are low
4. Methodology, line 355 and further - this paragraph does not describe measurement of Cd ion concentration as indicated in the heading
5. English language - the manuscript should be corrected for wording and grammar mistakes
Comments on the Quality of English Language
The manuscript should be corrected for wording and grammar mistakes
Author Response
Please find the rebuttal letter.
thanks.

Reviewer 2 Report
Comments and Suggestions for Authors
The authors present an investigation into the impact quantum dots have on receptor-mediated endocytosis and pinocytosis in yeast.
There are numerous major issues with the manuscript, the most prominent being the model utilised in this work (yeast), the extrapolation from this to mammalian cells, and the lack of information/justification surrounding the quantum dots used in the study.
There is no real description of the nanomaterials used in terms of surface functionalisation or stabilisation. Neither is there any real measure of the size or particle charge under the tested conditions (in biologically relevant media).
No consideration has been given to what the cells are actually "seeing" in terms of surface coatings or corona which, importantly, changes depending on the solution that the nanoparticles are in.
No justification has been provided for the tested concentrations. While predominantly observations were made via microscopy, any truly quantitative measures of e.g. cytotoxicity, would be far more impacted by this seemingly arbitrary choice.
While the yeast model is clearly established in the authors' laboratory, it does not pose a good surrogate for mammalian models which are discussed in the manuscript highlighting existing literature. Particularly pinocytosis as a mechanism for nanoparticle uptake by mammalian cells is largely irrelevant.
Specifically the sentence beginning line 275 is unsupported by the presented data, neither is the paragraph beginning line 302.
There are also numerous typos throughout which need to be addressed.
Comments on the Quality of English LanguageThere are numerous typos and spacing inconsistencies with references throughout the text.
Author Response
Please find the rebuttal letter.
Best,
Kyoungtae Kim

Round 2
Reviewer 1 Report
Comments and Suggestions for Authors
The Authors have responded to all my remarks and comments. In my opinion, the improved manuscript deserves publication in IJMS. Please, correct the references list in order to unify all the citations according to the journal's guidance.
Reviewer 2 Report
Comments and Suggestions for Authors
The proposed suggestions for improvement have been followed by the authors. I feel that these have suitably clarified key points of the manuscript and it is now in a publishable form.